# Association of Korean Healthy Eating Index and Sleep Duration with Obesity in Korean Adults: Based on the 7th Korea National Health and Nutrition Examination Survey 2016–2018

**DOI:** 10.3390/nu16060835

**Published:** 2024-03-14

**Authors:** Youngmin Namgung, Won Jang, Oran Kwon, Hyesook Kim

**Affiliations:** 1Department of Clinical Healthcare, Clinical Nutrition, Ewha Womans University, 52, Ewhayeodae-gil, Seodaemun-gu, Seoul 03760, Republic of Korea; ngym@ewhain.net; 2Department of Food and Nutrition, Wonkwang University, Iksan 54538, Republic of Korea; jangwon1011@naver.com; 3Institute for Better Living, Wonkwang University, Iksan 54538, Republic of Korea; 4Department of Nutritional Science and Food Management, Ewha Womans University, 52, Ewhayeodae-gil, Seodaemun-gu, Seoul 03760, Republic of Korea; orank@ewha.ac.kr; 5System Health & Engineering Major in Graduate School, Ewha Womans University, 52, Ewhayeodae-gil, Seodaemun-gu, Seoul 03760, Republic of Korea

**Keywords:** Korean Healthy Eating Index, diet quality, sleep duration, obesity, adults

## Abstract

Short sleep duration has been linked to an increased obesity risk, and emerging evidence suggests that diet quality potentially influences this association. This cross-sectional study aimed to examine the association of obesity with sleep duration and diet quality in adults. The participants comprised 10,967 adults (4623 men and 6344 women) aged 19–64 years who participated in the 7th National Health and Nutrition Examination Survey (2016–2018). Sleep duration was categorized into adequate (≥7 h) and insufficient (<7 h). Diet quality was evaluated using the Korean Healthy Eating Index (KHEI), with scores ranging from 0 to 100, based on 14 dietary components. Obesity was associated with higher rates of insufficient sleep in women but not in men. After adjusting for covariates, the obesity risk in women with insufficient sleep was approximately 1.3 times higher than that in women with adequate sleep (odds ratio [95% confidence interval] = 1.270 [1.058–1.525]), and this association was exclusively observed in the “KHEI ≤ median score” group (men, 59.95; women, 63.30). In conclusion, enhanced diet quality may act as an effect modifier in the association between insufficient sleep and a high obesity risk in women. These findings suggest that the association between sleep duration and obesity risk is potentially modified by dietary quality in adult women. Future studies with larger sample sizes and a prospective or interventional design are warranted to augment current knowledge regarding the association of diet quality/dietary patterns, and sleep duration with obesity.

## 1. Introduction

Obesity is becoming increasingly prevalent worldwide, and its associated health risks are also on the rise [1]. Numerous recent studies [2,3,4,5,6,7,8,9,10,11], a meta-analysis [12], and certain reviews [13,14,15] have yielded substantial evidence corroborating an association between short sleep duration and obesity. Moreover, recent research findings [9,10] indicate that insufficient sleep is linked to obesity owing to a decrease in the hormone leptin, which suppresses appetite, and an increase in the hormone ghrelin, which stimulates appetite. Furthermore, nocturnal serum levels of the stress hormone cortisol are elevated in individuals who obtain insufficient sleep. Cortisol tends to promote wakefulness and fat storage; thus, inadequate sleep potentially leads to increased fat storage and subsequent weight gain [16]. Insufficient sleep also culminates in heightened sympathetic nervous system activity and suppressed insulin secretion following intravenous glucose administration [17]. These physiological changes resulting from sleep deprivation are risk factors for insulin resistance [11], high blood pressure [6,7], and metabolic syndrome [5,8]. Moreover, individuals who obtain minimal sleep frequently experience reduced concentration owing to accumulated physical and mental fatigue. This, in turn, leads to decreased physical and mental activity, which can be linked to insufficient exercise and serve as another cause of weight gain [18]. Park et al. analyzed data from the Korea National Health and Nutrition Examination Survey (KNHANES) and proposed a significant association between short sleep duration and a moderate increase in both overall and abdominal obesity among Korean adults [19].

Substantial evidence indicates that diet quality potentially plays a crucial role in explaining the link between inadequate sleep duration and obesity. Several studies have revealed an association between insufficient sleep and increased food consumption, snacking, and poorer dietary quality [20,21,22]. Reduced sleep duration (<5 h) has been found to negatively affect body composition, macronutrient intake, and overall quality of life in individuals with obesity [23]. Additionally, increased sleep duration has been associated with reduced energy intake, a lower body mass index (BMI), and changes in the dietary intakes of saturated fatty acids (SFAs), polyunsaturated fatty acids, and carbohydrates by sex and age [24]. Shorter sleep duration is also linked to increased overweight and obesity risks, along with poorer dietary quality [25]. Another study observed that the association between sleep quality and obesity in Korean women was modified by the Recommended Food Score [26].

The Healthy Eating Index (HEI) was developed by the United States Department of Agriculture and the Department of Health and Human Services to comprehensively measure diet quality based on the Dietary Guidelines for Americans and the Food Guide Pyramid [27]. On this premise, the Korea Centers for Disease Control and Prevention developed the Korean Healthy Eating Index (KHEI) [28]. According to certain studies, the HEI is linked to lower risks of both overall and abdominal obesity [29,30] as well as sleep duration [31,32].

Dietary factors, especially overall dietary quality, possibly influence the association between sleep status and obesity [24,26,29]; nonetheless, to the best of our knowledge, no studies have examined the association of sleep duration and obesity with the KHEI, which reflects overall dietary patterns and eating habits. Therefore, this study aimed to determine whether the association between sleep duration and obesity varies according to the KHEI score using nationally representative data.

## 2. Materials and Methods

### 2.1. Participants

This study directly analyzed the 7th KNHANES (2016–2018) data. In brief, the KNHANES, a continuous cross-sectional survey of South Koreans across all ages, has been conducted by the Korea Centers for Disease Control (KCDC) since 1998. It employs multistage, stratified, probability-clustered sampling to select primary sampling units, that is, households belonging to non-institutional residents of South Korea. Furthermore, the KNHANES comprises a health interview, health examination, and nutritional survey and collects data on several variables pertaining to the participants’ demographic, social, health, and nutritional status. Detailed information on the KNHANES is electronically available (https://knhanes.kdca.go.kr/knhanes/eng/index.do, accessed on 12 March 2024). Written informed consent was obtained from all participants prior to survey enrollment. The KNHANES received approval from the Institutional Review Board (IRB) of the KCDC. The present study was conducted in accordance with the guidelines of the Declaration of Helsinki, and its protocols were approved by the IRB of Ewha Womans University (ewha-202211-0026-01).

Of the 24,269 KNHANES (2016–2018) participants, we included adults aged 19–64 years (*n* = 14,433) and excluded pregnant and lactating women (*n* = 163). Participants with (1) missing data on sleep duration (*n* = 1360), (2) missing BMI data (*n* = 21), (3) missing food intake data (*n* = 1841), and (4) an implausible energy intake (<500 or >8000 kcal/day; *n* = 81) were also excluded. Finally, 10,967 participants (4623 men and 6344 women) aged 19–64 years were included in this analysis. Based on the median KHEI by sex, participants were categorized into “KHEI ≤ median score” (2311 men and 3172 women) and “KHEI > median score” (2312 men and 3172 women) groups (Figure 1).

### 2.2. General and Socioeconomic Characteristics

All participants were interviewed by trained interviewers. This study used 7th KNHANES (2016–2018) data on demographic and socioeconomic characteristics: age, height, weight, BMI, education level, marital status, occupational status, household income, drinking status, smoking status, physical activity, dietary supplementation, and menopausal status (women only). Based on education level, participants were categorized into four groups: elementary school graduates, middle school graduates, high school graduates, and university graduates and above. Marital status was classified as single or married. Regarding occupational status, participants were classified as employed or unemployed. Household income was divided into low, middle-low, middle-high, and high quartiles. Drinking status was divided into five groups: never, less than once a month, 2–4 times a month, 2–3 times a week, and 4 or more times a week. Smoking status was determined based on the current scenario (current smoker, past smoker, or non-smoker). Regarding dietary supplementation or physical activity (aerobic), participants simply had to respond “yes” or “no”. For physical activity, they provided a “yes” response if they engaged in ≥150 min of moderate physical activity or ≥75 min of vigorous physical activity per week and a “no” response if otherwise.

### 2.3. Anthropometric Measurements

Height and weight were measured by trained staff, and BMI was calculated as follows: body weight divided by height squared (kg/m^2^). Obesity was defined as a BMI ≥ 25 kg/m^2^, according to the definitions of the World Health Organization Asia-Pacific region [33], the Korean Society of Obesity [34], and KNHANES. Obesity status was evaluated based on BMI, and no body composition tests were performed to determine percentage fat mass.

### 2.4. Assessment of Sleep Duration

Sleep duration data were collected using a self-reported questionnaire developed for the KNHANES via a health interview. Average sleep duration was calculated based on the following question: “On weekdays or workdays, what time do you go to sleep and wake up?” [35,36,37]. According to the National Sleep Foundation of the United States, 7–8 h of sleep is recommended for those who sleep < 6 h/day because of the increased risks of obesity and chronic disease; thus, sleep duration was categorized into short/inadequate (<7 h/day) and adequate (≥7 h/day) [38].

### 2.5. Assessment of Diet Quality

#### 2.5.1. Macronutrient Intake

Dietary intake was assessed via a 24-h dietary recall [39]. Data were collected from participants by KCDC-trained dietary interviewers via face-to-face interviews, wherein participants reported all the food and drinks consumed the previous day. Daily energy, macronutrient (carbohydrate, protein, and fat), and percentage macronutrient intakes were calculated.

#### 2.5.2. KHEI

We used the KHEI, which was developed by the KCDC based on KNHANES data [28]. KHEI components were selected based on the Dietary Guidelines for Koreans, domestic and overseas dietary quality indices, and the analysis results of the association of these components with chronic diseases. The KHEI score is based on the 2015 Dietary Reference Intakes for Koreans (KDRI). The KHEI score of Korean adults was calculated using KNHANES 24-h dietary recall data, and the maximum score is 100. It comprises 14 components clustered into three categories: (1) adequacy (eight components), which ascertains whether the recommended food and nutrients are consumed (breakfast consumption, mixed grain intake, total fruit intake, fresh fruit intake, total vegetable intake, vegetable intake excluding kimchi/pickled vegetable intake, meat/fish/egg/bean intake, and milk/milk product intake); (2) moderation (three components), which evaluates restricted food and nutrient intakes (percentage energy intake from SFAs, sodium intake, and percentage energy intake from sweets and beverages); and (3) energy balance (three components), which assesses energy balance and macronutrient intake (percentage energy intake from carbohydrates, percentage energy intake from fat, and overall energy intake) (Appendix A).

### 2.6. Statistical Analysis

Owing to the KNHANES study’s complex sampling design, relevant primary sampling units, stratification, and sample weights were exclusively considered in this analysis. Regarding descriptive statistics, continuous and categorical variables are expressed as the weighted mean ± standard error and numbers with weighted percentages using the SURVEYMEANS and SURVEYFREQ procedures, respectively. Significance was tested based on a BMI of 25 kg/m^2^ using the *t*-test and Rao-Scott chi-square test for continuous and categorical variables, respectively. To determine whether the association of sleep duration with obesity varies with diet quality (KHEI median), a SURVEY LOGISTIC analysis was performed, with age, education level, household income, marital status, occupational status, smoking status, drinking status, physical activity, dietary supplementation, and menopausal status (women only) as covariates. The odds ratios (ORs) and 95% confidence intervals (CIs) for obesity were estimated in correspondence to a sleep duration of ≥7 h per day. All data were statistically processed using the SAS program (SAS 9.4; SAS Institute, Cary, NC, USA), and statistical significance was assigned at a *p*-value less than 0.05.

## 3. Results

### 3.1. General Characteristics of the Participants According to BMI Status

This study analyzed the data of 10,967 adults (4623 men and 6344 women). The study participants’ general characteristics by sex and BMI are displayed in Table 1. The obesity numbers for men and women were 2015 and 1662, accounting for 18.4% and 15.2% of all participants, respectively. Women with obesity were generally older and displayed higher married and postmenopausal rates than those without obesity (*p* < 0.001), and they also tended to have a lower employment rate (*p* = 0.048). On comparing general characteristics by sex, men were generally younger, had higher BMI values, exhibited greater physical activity, and had lower dietary supplement use than women (*p* < 0.001). Socioeconomic characteristics and health-related behaviors, such as education level (*p* < 0.001), marital status (*p* < 0.001), occupational status (*p* < 0.001), household income (*p* = 0.007), smoking status (*p* < 0.001), and drinking status (*p* < 0.001), were significantly different between men and women.

### 3.2. Sleep Duration According to BMI Status

Table 2 shows the sleep duration results according to obesity status. In terms of average sleep duration, women slept longer than men (*p* = 0.012). The average sleep duration of women with obesity was shorter than that of those without (*p* = 0.003), and a significant difference in insufficient/short sleep duration (<7 h/day) was noted (*p* = 0.003). In contrast, men exhibited no significant differences in both average and insufficient/short sleep duration (<7 h/day) according to obesity status.

### 3.3. Diet Quality (KHEI) According to BMI Status

Table 3 lists the KHEI scores according to BMI. The diet quality (KHEI) of women with obesity was lower than that of those without (*p* = 0.009). In contrast, the diet quality of men displayed no significant difference according to obesity status. However, among the KHEI components, fresh fruit (*p* = 0.038), total vegetable (*p* = 0.009), vegetable (excluding kimchi and pickled vegetables) (*p* = 0.014), and meat/fish/egg/bean (*p* = 0.002) intakes were lower in men with obesity than in those without.

In women with obesity, the following KHEI components yielded significantly lower values than those in women without obesity: meat/fish/egg/bean intake (*p* = 0.002); adequate milk/milk product intake (*p* < 0.001); percentage energy intake from SFA (*p* = 0.006) in moderation; and energy balance in terms of the percentage energy intake from carbohydrates (*p* = 0.001), protein (*p* = 0.004), and fat (*p* < 0.001).

A significant difference in diet quality (KHEI) was observed between men and women; women yielded a higher KHEI score than men (*p* < 0.001). According to sex, adequacy in terms of breakfast consumption (*p* = 0.001), mixed grain intake (*p* = 0.022), total fruit intake (*p* < 0.001), fresh fruit intake (*p* < 0.001), total vegetable intake (*p* < 0.001), vegetable intake (excluding kimchi and pickled vegetables) (*p* < 0.001), meat/fish/egg/bean intake (*p* < 0.001), and milk/milk product intake (*p* < 0.001) exhibited significant differences. Significant differences were also noted in moderation regarding sodium intake (*p* < 0.001) and energy balance in terms of the percentage energy intake from carbohydrates (*p* = 0.047).

### 3.4. Macronutrient Intake According to the Median KHEI Score

Table 4 shows the macronutrient and percentage energy intakes according to diet quality (KHEI median value). Macronutrient intake in men displayed a significant difference according to diet quality (KHEI). Men with a KHEI score below the median value had a lower carbohydrate intake (*p* < 0.001), lower protein intake (*p* = 0.001), and higher fat intake (*p* < 0.001) than those with a KHEI score above the median value (*p* < 0.001). Furthermore, in men with a KHEI score below the median value, carbohydrate (*p* < 0.001) and fat (*p* < 0.001) contributions to energy intake were lower and higher, respectively, than those in men with a KHEI score above the median value.

Women with a KHEI score below the median value had a lower overall energy intake than those with a KHEI score above the median value. Moreover, women with a KHEI score below the median value had a lower carbohydrate intake (*p* < 0.001), lower protein intake (*p* = 0.001), and higher fat intake (*p* = 0.020) than those with a KHEI score above the median value (*p* < 0.001). Furthermore, in women with a KHEI score below the median value, carbohydrate (*p* < 0.001) and protein (*p* < 0.001) contributions to energy intake were lower, whereas that of fat (*p* < 0.001) was higher than those in women with a KHEI score above the median value.

### 3.5. Association between Sleep Duration and Obesity According to the Median KHEI Score

The data in Table 5 were analyzed using a multiple logistic regression model to evaluate the association between sleep duration and obesity according to the median KHEI score. In women with a KHEI score ≤ median value, short sleep duration (<7 h/day) was associated with a higher obesity risk than adequate sleep duration (≥7 h/day) (OR [95% CI] = 1.349 [1.134–1.605]) in the unadjusted model. Even after the model had been adjusted for age, education level, household income, marital status, occupational status, smoking status, drinking status, aerobic physical activity, supplement intake, and menopausal status (women only), the results remained the same. In women with a KHEI score > median value, short sleep duration (<7 h/day) was associated with a higher obesity risk than adequate sleep duration (≥7 h/day) (adjusted OR [95% CI] = 1.270 [1.058–1.525]). These results revealed consistent associations in other age groups, except in women aged 19–34 years, women aged 35–49 years with a KHEI score ≤ median value (OR [95% CI] = 1.553 [1.150–2.095] and adjusted OR [95% CI] = 1.455 [1.059–2.000]), and women aged 50–64 years with a KHEI score ≤ median value (OR [95% CI] = 1.405 [1.070–1.846] and adjusted OR [95% CI] = 1.459 [1.103–1.930]). In men, no significant association was observed between sleep duration and obesity in the unadjusted and adjusted models.

## 4. Discussion

This study performed a large-scale epidemiological analysis of Korean adults aged 19–64 years to assess the association of obesity with sleep duration and diet quality. Women who slept < 7 h/day were found to carry approximately 1.5 times the risk of obesity compared with those who slept ≥ 7 h/day, and this association was exclusively identified in women with a KHEI score below the median value (men: 59.95 and women: 63.3). These results suggest that dietary quality is a potential modifying factor in the association between sleep duration and obesity risk in adult women.

Previous studies that have used the HEI focused on examining the association between diet quality and obesity [29,30,40] or diet quality and sleep duration [31,32,36,41,42]. However, only two studies have investigated the association among sleep duration, obesity, and dietary intake in Korean adults [26,43]. We found a significant difference in the KHEI score between men and women; women had a higher diet quality (KHEI score) than men. Differences in diet quality according to obesity status were noted in women; the KHEI score was lower in women with obesity than in those without. Several previous studies have suggested a significant association between the HEI and obesity. Among them, a study based on data from the National Health and Nutrition Examination Survey (NHANES) III, which targeted adults aged 20–75 years in the United States, reported that a low HEI score was strongly associated with overweight and obesity [24]. In addition, a cross-sectional study of NHANES III data found the total HEI score and HEI component scores to be significantly associated with abdominal adiposity in adults aged ≥ 20 years [30]. Moreover, another study using cohort data from the Multi-Ethnic Study of Atherosclerosis in the United States indicated that obesity could be predicted by the HEI score according to race [40].

In the present study, no significant difference in diet quality (KHEI score) was observed between men and women according to sleep duration. Another study also failed to identify a significant association between sleep duration and diet quality [36,44]. However, certain studies have reported a significant association between the HEI and sleep duration. For instance, short sleep duration has been associated with lower HEI scores in adults [31], poorer dietary quality in Hispanic/Latino adults [45], and a lower rate of diet quality interruption in postmenopausal women [32]. In addition, young women with short sleep durations were found to have significantly lower scores across various diet quality indices [25].

On analyzing average sleep duration according to obesity, we found women to have a significantly longer average sleep duration than men (Table 2). The average sleep duration of women with obesity was shorter than that of those without, and a significant difference in insufficient/short sleep duration (<7 h/day) was also noted. However, in men, no significant difference was observed in both average sleep duration and insufficient/short sleep duration (<7 h/day) according to obesity. The cause of these results is unknown, as the exact mechanism underlying the relationship between sleep duration and obesity is yet to be elucidated.

However, as mentioned above, short sleep duration potentially increases obesity risk by affecting hormones, such as leptin and ghrelin, and altering dietary intake. A study investigating the association between sleep duration (using polysomnography) and BMI in participants from the Wisconsin Sleep Cohort found sleep duration to decrease with increasing BMI in the groups sleeping less than 8 h; moreover, leptin and ghrelin levels decreased and increased, respectively, in the short sleep duration groups [19]. This difference in activity between leptin and ghrelin has the potential to increase appetite and possibly explain the increase in BMI observed with short sleep duration [9]. In a cross-sectional study examining the changes in adult sleep duration, BMI, and leptin levels among participants from the Québec Family Study, lower adiposity indices were noted in both men and women in the “7–8-h sleep” group than in the “5–6-h sleep” group. These results indicate an association of short sleep duration with weight gain and obesity; furthermore, as sleep duration is a hypothetically modifiable risk factor, these findings suggest important clinical implications for obesity prevention and treatment [10]. In a small clinical trial involving American adult men, sleep duration restriction decreased the anorexigenic hormone leptin and increased the appetite-inducing hormone ghrelin; additionally, elevated appetite and hunger for high-calorie foods were correlated with high carbohydrate content [46].

In a large-scale epidemiological study of American adults aged 18–65 years, the probability of developing obesity significantly increased in adults who slept 5–6 or <5 h compared with that in adults who slept 7–8 h [47]. A prospective cohort study involving 27-year-olds in Switzerland conducted over a 13-year period found that BMI tended to be lower among adults who slept longer, and a significant association was noted between shorter sleep duration and obesity [48]. In addition, the association between short sleep duration (<6 h) and obesity weakened with age, and sleep duration was more strongly associated with BMI before the survey than during the study or at follow-up [49].

Several reviews and meta-analyses have confirmed the overall association of sleep duration with obesity [13,48,49,50,51]. In a systematic literature review of 36 studies (31 cross-sectional, 5 prospective) reporting an association between short sleep duration and at least one measure of weight, Patel and Hu [13] established a strong and persistent association between short sleep duration and weight gain in children. Nonetheless, adult analyses yielded mixed results, with 17 out of 23 studies reporting an independent relationship between short sleep duration and increased weight and all three longitudinal studies corroborating a positive association between short sleep duration and future weight; nevertheless, this relationship tended to disappear with age [13]. However, major study design constraints prevented definitive conclusions. In the same year (2008), a critical review of the association between sleep duration and obesity in adults and children concluded that while some cross-sectional and longitudinal studies have demonstrated an association between short sleep duration and BMI in adults, the evidence base was insufficient to provide the general population or specific groups with public health recommendations regarding the applicability of sleep duration as a modifiable risk factor for obesity [49]. A meta-analysis of cross-sectional studies, also published in 2008, reported a persistently increased obesity risk among children and adults with short sleep durations [50]. According to a meta-analysis updated in 2014, short sleep duration was significantly associated with the occurrence of obesity, while long sleep duration exerted no effect on future obesity in adults [12].

In this study, an association between sleep duration and obesity was exclusively observed in women with a KHEI score ≤ median value. As mentioned above, the KHEI was developed based on the KDRI and dietary guidelines and is a potentially useful tool in assessing the association between overall dietary quality and chronic disease. This possibly implies that individuals with a KHEI score ≤ median value tend not to follow the KDRI and dietary guidelines. As shown in Table 4, regarding macronutrient intake, women with a KHEI score ≤ median value exhibited lower carbohydrate and protein intakes but higher fat intake than those with a KHEI score > median value. In addition, although it was not analyzed in this study, another study observed that micronutrient intake in adults with short sleep duration was significantly lower than the estimated average requirement, and inadequate intake differed according to sex [52]. Furthermore, another study reported a significant difference in micronutrient intake between participants with short and long sleep durations [53].

This study yielded different associations according to sex. These variations between men and women can be attributed to hormonal differences. Sleep intervention studies have found hormonal changes to differ between men and women, suggesting that they are vulnerable to overeating via separate mechanisms [54]. In a study wherein men and women were equally restricted from sleep while provided free access to food, morning plasma leptin levels increased significantly, especially in women [55]. An examination of sex hormones and obesity suggested that different hormonal influences in men and women are related to obesity [56]. A previous study found the association of sleep duration with obesity in adults to be stronger in women than in men [57].

In addition, as depicted in Table 5, consistent associations were displayed in other age groups, except in participants aged 19–34 years. The association between sleep duration and BMI varies with age. In young adults, the association between sleep duration and BMI tends to change after middle age [58], possibly explaining this study’s results. These results potentially indicate that the association between insufficient sleep duration and obesity risk may not appear in women with a high KHEI score, an index that reflects the overall dietary quality of Korean adults.

This study has certain limitations that require further discussion. First, because of the cross-sectional nature of the KNHANES, we could not establish causality in the association between sleep duration and obesity owing to diet quality. Second, in this study, the KHEI was used to evaluate the diet quality of subjects, and only the percentage of macronutrients was examined. No tools were used to evaluate other diets. In addition, in this study, the KHEI was divided into categorical variables rather than mean or continuous variables. The importance of exploring alternative classifications or treating KHEI as a continuous variable in future studies may further reveal the complex relationship between dietary quality, sleep duration, and obesity. Third, although demographic and socioeconomic characteristics were considered in this study, unmeasured confounding variables might have existed, for example, sedentary behavior, depression, and drug use. Fourth, to evaluate sleep duration, the KNHANES self-questionnaire was used. In practice, sleep duration should also consider nap time, sleep quality, bedtime, and sleep disorders (chronic insomnia and obstructive sleep apnea syndrome). However, in a large-scale census, the detailed measurement of sleep duration is encumbered by practical difficulties. In addition, each participant’s sleep duration was recorded via recall, and recalling sleep duration accurately may not be possible; therefore, the results of the studied association may be biased. We selected sleeping duration during weekdays or workdays, as this study was based on sleep duration data from the KNHANES and previous overseas studies [35,36,37]. Sleep duration was classified as <7 and ≥7 h; it was not further subdivided, and this analysis did not examine excessive sleep. There are studies reporting that the longer the sleep duration, the more U-shaped the gap between sleep duration and obesity [9,48]. Consequently, our study is susceptible to potential bias and data noise introduced by the inclusion of participants with excessive sleep. To enhance the validity of our conclusions, future studies should consider incorporating additional categories of sleep duration, such as long sleep duration. We propose further investigations to provide a more nuanced understanding of the relationship between sleep duration and its potential impacts. Fifth, in this study, obesity was defined solely based on BMI. Body fat percentage is a potentially more accurate indicator of obesity. However, the KNHANES does not conduct body composition evaluations; therefore, we were unable to consider this indicator. Notwithstanding, BMI remains a widely used tool for determining obesity.

Despite these limitations, this study is the first to investigate the association of sleep duration with obesity according to the KHEI score, which indicates diet quality in Koreans, in a representative population sample using national-scale data. Obesity is a complex disorder with multifactorial causes, and this study is meaningful in that it presents diet quality as a modifiable factor influencing sleep duration that should not be overlooked in obesity prevention and intervention efforts. Further investigations into specific dietary factors that help mitigate obesity risk in the context of sleep deprivation are warranted.

## 5. Conclusions

Our findings reveal that women with insufficient sleep have a significantly higher obesity risk, especially when their KHEI score, which indicates their diet quality, is low. They also suggest that an eating pattern that meets dietary reference intakes and dietary guidelines contributes to an important and meaningful healthy lifestyle that reduces the obesity risk caused by a lack of sleep. Future studies involving larger sample sizes and a prospective or interventional design are required to enhance current knowledge regarding the association of diet quality/dietary patterns, and sleep duration with obesity.

## Figures and Tables

**Figure 1 nutrients-16-00835-f001:**
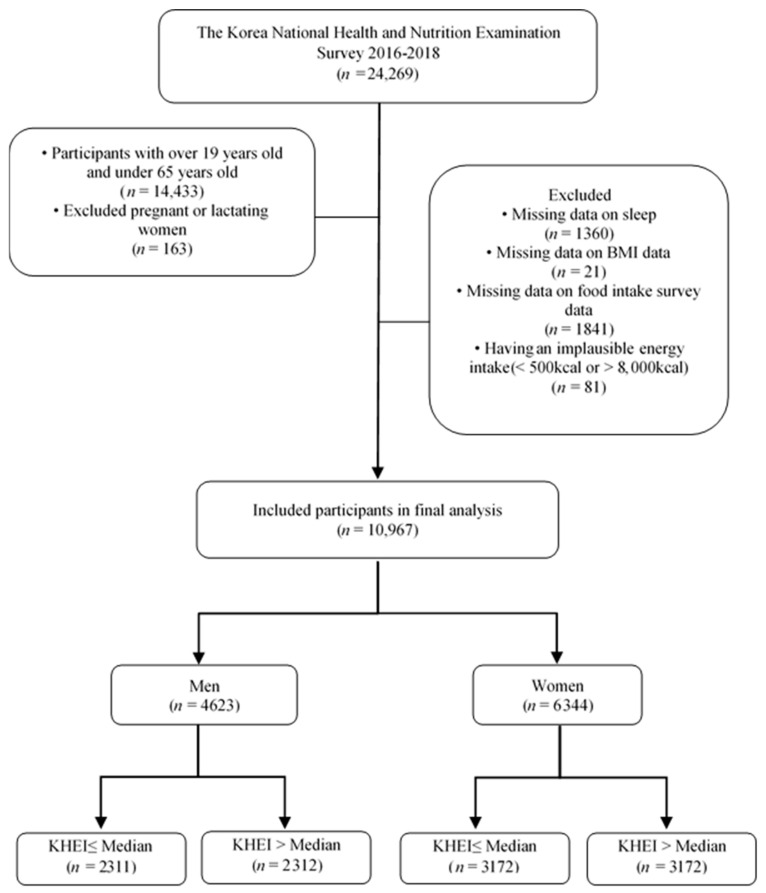
Flow diagram of participant enrollment.

**Table 1 nutrients-16-00835-t001:** General characteristics of the subjects by BMI.

	Men				Women				*p*-Value **
	Total (*n* = 4623)	BMI < 25 kg/m^2^ (*n* = 2608)	BMI ≥ 25 kg/m^2^ (*n* = 2015)	*p*-Value *	Total (*n* = 6344)	BMI < 25 kg/m^2^ (*n* = 4682)	BMI ≥ 25 kg/m^2^ (*n* = 1662)	*p*-Value *	
Age (years)	41.28 ± 0.24 ^1^	41.15 ± 0.31	41.43 ± 0.32	0.482	42.55 ± 0.21	41.53 ± 0.24	45.66 ± 0.37	<0.001	<0.001
Height (cm)	172.54 ± 0.11	172.62 ± 0.14	172.43 ± 0.16	0.367	159.43 ± 0.09	159.77 ± 0.10	158.39 ± 0.18	<0.001	<0.001
Weight (kg)	73.70 ± 0.19	66.66 ± 0.18	82.90 ± 0.26	<0.001	58.49 ± 0.15	54.55 ± 0.10	70.52 ± 0.28	<0.001	<0.001
BMI (kg/m^2^)	24.72 ± 0.06	22.35 ± 0.05	27.83 ± 0.07	<0.001	23.02 ± 0.06	21.37 ± 0.04	28.06 ± 0.08	<0.001	<0.001
Education level									
≤Elementary	279 (4.24)	183 (4.90)	96 (3.37)	0.011	590 (7.29)	309 (5.18)	281 (13.74)	<0.001	<0.001
≤Middle school	355 (5.86)	212 (5.74)	143 (6.02)		581 (8.17)	380 (7.08)	201 (11.48)		
≥High school	1694 (38.63)	972 (39.87)	722 (37.01)		2355 (38.56)	1709 (37.73)	646 (41.10)		
≥College	2292 (51.28)	1239 (49.49)	1053 (53.61)		2816 (45.98)	2283 (50.00)	533 (33.68)		
Marital status									
Married	3357 (66.18)	1839 (63.87)	1518 (69.20)	0.001	5242 (77.10)	3776 (74.81)	1466 (84.10)	<0.001	<0.001
Single	1266 (33.82)	769 (36.13)	497 (30.80)		1102 (22.90)	906 (25.19)	196 (15.90)		
Occupation									
Yes	3789 (81.12)	2102 (79.85)	1687 (82.80)	0.035	3863 (60.00)	2887 (60.81)	976 (57.53)	0.048	<0.001
No	820 (18.88)	500 (20.15)	320 (17.20)		2478 (40.00)	1792 (39.19)	686 (42.47)		
Household income									
Low	432 (8.97)	264 (9.52)	168 (8.24)	0.430	601 (9.08)	383 (8.12)	218 (12.04)	<0.001	0.007
Middle-low	1017 (21.24)	580 (21.49)	437 (20.92)		1553 (23.94)	1064 (22.12)	489 (29.53)		
Middle-high	1457 (31.89)	794 (31.04)	663 (32.99)		1980 (31.57)	1456 (31.16)	524 (32.81)		
High	1712 (37.91)	967 (37.95)	745 (37.85)		2202 (35.40)	1773 (38.60)	429 (25.63)		
Drinking									
None	586 (11.71)	344 (12.51)	242 (10.67)	0.069	1642 (23.11)	1150 (21.91)	492 (26.77)	0.005	<0.001
≤1 drink/month	1034 (23.89)	592 (24.09)	442 (23.62)		2281(36.49)	1697 (36.88)	584 (35.32)		
2–4 drinks/month	1303 (29.30)	747 (30.14)	556 (28.20)		1531 (25.77)	1180 (26.58)	351 (23.30)		
2–3 drinks/week	1183 (25.33)	644 (23.94)	539 (27.13)		711 (11.65)	531 (11.80)	180 (11.17)		
≥4 drinks/week	513 (9.77)	279 (9.31)	234 (10.38)		173 (2.98)	119 (2.83)	54 (3.44)		
Smoking									
Non-smoker	1180 (27.79)	691 (29.10)	489 (26.07)	0.095	5557 (86.82)	4123 (87.08)	1434 (86.00)	0.602	<0.001
Past smoker	1656 (33.78)	916 (32.63)	740 (35.28)		410 (6.60)	293 (6.52)	117 (6.83)		
Current smoker	1781 (38.43)	998 (38.26)	783 (38.65)		368 (6.59)	260 (6.39)	108 (7.17)		
Physical activity									
Yes	2297 (52.76)	1276 (52.07)	1021 (53.66)	0.327	2894 (48.43)	2191 (49.12)	703 (46.31)	0.080	<0.001
No	2323 (47.24)	1332 (47.93)	991 (46.34)		3444 (51.57)	2487 (50.88)	957 (53.69)		
Dietary supplement									
Yes	2127 (45.05)	1238 (46.44)	889 (43.25)	0.058	3494 (53.48)	2617 (53.69)	877 (52.81)	0.596	<0.001
No	2496 (54.95)	1370 (53.56)	1126 (56.75)		2850 (46.52)	2065 (46.31)	785 (47.19)		
Postmenopausal									
Yes					2299 (31.62)	1555 (28.96)	744 (39.75)	<0.001	
No					4045 (68.38)	3127 (71.04)	918 (60.25)		

^1^ The data were expressed as means ± S.E (standard error) or frequency (%). Statistics are weighted using the individual weights provided by the KNHANES study participants. * *p*-value obtained by Body Mass Index (BMI) 25 kg/m^2^ using *t*-test for continuous variables or chi-square tests for categorical variables. ** *p*-value obtained by sex using *t*-test for continuous variables or chi-square tests for categorical variables.

**Table 2 nutrients-16-00835-t002:** Sleep duration of the subjects by BMI.

	Men				Women				*p*-Value **
	Total (*n* = 4623)	BMI < 25 kg/m^2^ (*n* = 2608)	BMI ≥ 25 kg/m^2^ (*n* = 2015)	*p*-Value *	Total (*n* = 6344)	BMI < 25 kg/m^2^ (*n* = 4682)	BMI ≥ 25 kg/m^2^ (*n* = 1662)	*p*-Value *	
Average sleep duration (h/day)	6.91 ± 0.02 ^1^	6.95 ± 0.03	6.87 ± 0.03	0.070	6.99 ± 0.02	7.02 ± 0.02	6.89 ± 0.04	0.003	0.012
Sleep duration < 7 h/day	1898 (41.48)	1039 (40.50)	859 (42.77)	0.175	2458 (40.29)	1758 (39.11)	700 (43.92)	0.003	0.279
Sleep duration ≥ 7 h/day	2725 (58.52)	1569 (59.50)	1156 (57.23)		3886 (59.71)	2924 (60.89)	962 (56.09)		

^1^ The data were expressed as means ± S.E (standard error) or frequency (%). * *p*-value obtained by Body Mass Index (BMI) 25 kg/m^2^ using *t*-test for continuous variables or chi-square tests for categorical variables. ** *p*-value obtained by sex using *t*-test for continuous variables or chi-square tests for categorical variables.

**Table 3 nutrients-16-00835-t003:** KHEI score of the subjects by BMI.

Component of KHEI Score	Men				Women				*p*-Value **
	Total (*n* = 4623)	BMI < 25 kg/m^2^ (*n* = 2608)	BMI ≥ 25 kg/m^2^ (*n* = 2015)	*p*-Value *	Total (*n* = 6344)	BMI < 25 kg/m^2^ (*n* = 4682)	BMI ≥ 25 kg/m^2^ (*n* = 1662)	*p*-Value *	
Total KHEI score (0–100)	59.96 ± 0.25 ^1^	59.99 ± 0.33	59.92 ± 0.34	0.871	63.30 ± 0.22	63.59 ± 0.26	62.42 ± 0.39	0.009	<0.001
Adequacy (8)									
Have breakfast (0–10)	6.46 ± 0.08	6.51 ± 0.10	6.40 ± 0.10	0.423	6.77 ± 0.07	6.72 ± 0.08	6.94 ± 0.12	0.100	0.001
Mixed grains intake (0–5)	1.73 ± 0.04	1.72 ± 0.05	1.73 ± 0.05	0.906	1.82 ± 0.03	1.81 ± 0.04	1.87 ± 0.06	0.350	0.022
Total fruit intake (0–5)	1.57 ± 0.03	1.63 ± 0.05	1.49 ± 0.05	0.038	2.43 ± 0.03	2.45 ± 0.04	2.36 ± 0.07	0.202	<0.001
Fresh fruit intake (0–5)	1.78 ± 0.04	1.82 ± 0.05	1.71 ± 0.06	0.127	2.64 ± 0.04	2.66 ± 0.04	2.57 ± 0.07	0.206	<0.001
Total vegetable intake (0–5)	3.70 ± 0.02	3.65 ± 0.03	3.77 ± 0.03	0.009	3.13 ± 0.02	3.10 ± 0.03	3.20 ± 0.04	0.073	<0.001
Vegetable intake, excluding kimchi and pickled vegetable intake (0–5)	3.31 ± 0.03	3.25 ± 0.04	3.38 ± 0.04	0.014	2.93 ± 0.02	2.93 ± 0.03	2.93 ± 0.05	0.999	<0.001
Meat/fish/eggs/beans intake (0–5)	7.58 ± 0.05	7.45 ± 0.07	7.74 ± 0.07	0.002	6.96 ± 0.05	7.05 ± 0.05	6.70 ± 0.10	0.002	<0.001
Milk/milk products intake (0–5)	3.25 ± 0.08	3.30 ± 0.10	3.20 ± 0.11	0.476	3.76 ± 0.07	3.93 ± 0.08	3.25 ±0.12	<0.001	<0.001
Moderation (3)									
Percentage of energy from saturated fatty acid (0–10)	6.80 ± 0.07	6.84 ± 0.10	6.74 ± 0.11	0.509	6.94 ± 0.07	6.84 ± 0.07	7.22 ± 0.13	0.006	0.133
Sodium intake (0–10)	5.42 ± 0.06	5.50 ± 0.08	5.31 ± 0.09	0.096	7.61 ± 0.04	7.62 ± 0.05	7.56 ± 0.09	0.519	<0.001
Percentage of energy from sweets and beverages (0–10)	9.04 ± 0.04	9.03 ± 0.06	9.06 ± 0.07	0.730	9.06 ± 0.04	9.05 ± 0.04	9.12 ± 0.07	0.414	0.639
Energy balance (3)									
Percentage of energy from carbohydrate (0–5)	2.77 ± 0.04	2.73 ± 0.05	2.83 ± 0.05	0.144	2.68 ± 0.03	2.74 ± 0.03	2.51 ± 0.06	0.001	0.047
Percentage of energy intake from fat (0–5)	3.55 ± 0.03	3.53 ± 0.04	3.59 ± 0.05	0.396	3.50 ± 0.03	3.55 ± 0.03	3.36 ± 0.06	0.004	0.215
Energy intake (0–5)	3.01 ± 0.04	3.04 ± 0.05	2.96 ± 0.06	0.265	3.06 ± 0.03	3.13 ± 0.04	2.85 ± 0.06	<0.001	0.274
KHEI ≤ median (%) ^2^	2311 (52.70)	1313 (53.04)	998 (52.25)	0.621	3172 (52.47)	2273 (51.31)	899 (56.02)	0.005	0.829

^1^ The data were expressed as means ± S.E (standard error) or frequency (%). ^2^ Korean Healthy Eating Index (KHEI) median values are 59.95 in men and 63.30 in women. * *p*-value obtained by Body Mass Index (BMI) 25 kg/m^2^ using *t*-test for continuous variables or chi-square tests for categorical variables. ** *p*-value obtained by sex using *t*-test for continuous variables or chi-square tests for categorical variables.

**Table 4 nutrients-16-00835-t004:** Macronutrient intake of the subjects by KHEI median.

	Men				Women				*p*-Value **
	Total (*n* = 4623)	KHEI ≤ Median ^†^ (*n* = 2311)	KHEI > Median (*n* = 2312)	*p*-Value *	Total (*n* = 6344)	KHE I ≤ Median (*n* = 3172)	KHEI > Median (*n* = 3172)	*p*-Value *	
Energy intake (kcal)/day ^2^	2467.66 ± 18.02 ^1^	2464.27 ± 28.84	2471.44 ± 18.85	0.830	1735.36 ± 10.77	1621.22 ± 16.34	1861.36 ± 11.44	<0.001	<0.001
Percentage of macronutrient intake (kcal) (%)/day ^3^									
Percentage of energy from carbohydrate (%)/day	61.41± 0.20	59.36 ± 0.34	63.70 ± 0.21	<0.001	63.29 ± 0.17	62.01 ± 0.28	64.70 ± 0.19	<0.001	<0.001
Percentage of energy from protein (%)/day	16.05 ± 0.09	15.91 ± 0.13	16.22 ± 0.11	0.070	15.07 ± 0.07	14.71 ± 0.11	15.45 ± 0.09	<0.001	<0.001
Percentage of energy from fat (%)/day	22.53 ± 0.16	24.73 ± 0.27	20.09 ± 0.16	<0.001	21.65 ± 0.14	23.28 ± 0.23	19.84 ± 0.14	<0.001	<0.001
Macronutrient intake (kcal)/day									
Carbohydrate intake (kcal)/day	1356.07 ± 9.47	1249.59 ± 13.40	1474.71 ± 12.09	<0.001	1057.51 ± 6.83	940.13 ± 9.10	1187.07 ± 8.39	<0.001	<0.001
Protein intake (kcal)/day	362.03 ± 3.17	351.51 ± 5.11	373.75 ± 3.62	0.001	252.91 ± 1.95	228.41 ± 2.82	279.94 ± 2.21	<0.001	<0.001
Fat intake (kcal)/day	532.17 ± 6.29	585.10 ± 10.82	473.21 ± 5.60	<0.001	375.24 ± 3.76	383.49 ± 6.45	366.14 ± 3.63	0.020	<0.001

^1^ The data were expressed as means ± S.E (standard error) or frequency (%). ^2^ Energy intake (kcal)/day is the total daily energy intake. ^3^ Percentage of macronutrient intake (kcal) (%)/day is a ratio according to the total energy intake of carbohydrates, proteins, and fats. ^†^ The median values of the Korean Healthy Eating Index (KHEI) are 60.37 in men and 63.80 in women. * *p*-value obtained by KHEI median (60.37 in men and 63.80 in women) using *t*-test for continuous variables or chi-square tests for categorical variables. ** *p*-value obtained by sex using *t*-test for continuous variables or chi-square tests for categorical variables.

**Table 5 nutrients-16-00835-t005:** Adjusted odds ratios (OR) and 95% confidence intervals (CI) of obesity according to sleep duration by KHEI median.

	All ^1^ (*n* = 10,967)		19–34 Years ^2^ (*n* = 2622)		35–49 Years ^3^ (*n* = 4099)		50–64 Years ^4^ (*n* = 4246)	
	≤Median	>Median	≤Median	>Median	≤Median	>Median	≤Median	>Median
	OR (95% CI) *	OR (95% CI)	OR (95% CI)	OR (95% CI)	OR (95% CI)	OR (95% CI)	OR (95% CI)	OR (95% CI)
Men								
Unadjusted	1.024 (0.841–1.246)	1.183 (0.984–1.423)	0.820 (0.565–1.189)	1.107 (0.788–1.555)	0.926 (0.673–1.273)	1.335 (0.999–1.784)	1.318 (0.958–1.813)	1.124 (0.824–1.531)
Model 1	1.015 (0.831–1.240)	1.206 (0.999–1.457)	0.812 (0.553–1.191)	1.280 (0.892–1.836)	0.943 (0.682–1.304)	1.306 (0.968–1.764)	1.272 (0.918–1.762)	1.172 (0.847–1.621)
Women								
Unadjusted	1.349 (1.134–1.605)	1.087 (0.887–1.331)	0.791 (0.503–1.245)	1.346 (0.862–2.102)	1.553 (1.150–2.095)	0.860 (0.640–1.156)	1.405 (1.070–1.846)	1.007 (0.746–1.361)
Model 1	1.270 (1.058–1.525)	1.032 (0.835–1.275)	0.840 (0.523–1.350)	1.410 (0.883–2.252)	1.455 (1.059–2.000)	0.777 (0.563–1.071)	1.459 (1.103–1.930)	0.988 (0.717–1.362)

^1^ All Korean Healthy Eating Index (KHEI) median values are 60.37 in men and 63.80 in women. ^2^ 19–34 years KHEI median values are 54.98 in men and 56.87 in women. ^3^ 35–49 years KHEI median values are 60.23 in men and 64.62 in women. ^4^ 50–64 years KHEI median values are 65.28 in men and 69.08 in women. * ORs (95% CI) were calculated with reference to adequate sleep duration (≥7 h/day) using a multiple logistic regression model. Model 1 was adjusted for age, education level, household income, marital status, occupation, smoking, drinking, physical activity, dietary supplements, and menopause status (women only).

## Data Availability

The datasets analyzed in the current study are available from the corresponding author upon reasonable request.

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
