# Peer review of "Association of Korean Healthy Eating Index and Sleep Duration with Obesity in Korean Adults: Based on the 7th Korea National Health and Nutrition Examination Survey 2016–2018"

_nutrients, 2024, doi:10.3390/nu16060835_

Round 1

Reviewer 1 Report

Comments and Suggestions for Authors

The paper by Dr Namgung et al. reports the results of a cross-sectional study investigating the association of sleep duration, obesity and diet quality in adults using data from the 2016−2018 Korean National Health and Nutrition Examination Survey (KNHANES). The manuscript discusses an interesting topic of rising concern; however, some improvements are suggested

Major comments

  1. More categories of sleep duration are welcome in order to have more valid conclusions.
  2. The discussion section should be condensed, with a stronger emphasis on the clinical implications of the study.

Author Response

We sincerely appreciate the reviewer's insightful and constructive comments and suggestions. Please see our detailed responses below.

1.More categories of sleep duration are welcome in order to have more valid conclusions.

Reply) Thank you for your thoughtful review of our manuscript. Unfortunately, our study focused only on the short sleep time (less than 7 h) and acknowledged the limitations regarding the absence of analyses on individuals who sleep more than the recommended amount. As suggested, considering the potential effects of long sleep on factors such as obesity may be worth a more comprehensive understanding. We appropriately noted the importance of this consideration and recognized the need for follow-up studies to explore the effects of long sleep time on various outcomes. It is acknowledged in some studies that longer sleep time is associated with a U-shaped relationship with obesity. We understand that not investigating excessive sleep is a limitation of our study, as it can introduce bias and noise in the data, especially in the context of potential side effects associated with extended sleep time. This limitation was explicitly addressed in the Discussion section of the paper as follows:

"Sleep duration was classified as < 7 and ³ 7 h; it was not further subdivided, and this analysis did not examine excessive sleep. There are studies reporting that the longer the sleep duration, the more U-shaped the gap between sleep duration and obesity [9,48]. Consequently, our study is susceptible to potential bias and data noise introduced by the inclusion of participants with excessive sleep. To enhance the validity of our conclusions, future studies should consider incorporating additional categories of sleep duration, such as long sleep duration. We propose further investigations to provide a more nuanced understanding of the relationship between sleep duration and its potential impacts." (P5, L417–L425)

2. The discussion section should be condensed, with a stronger emphasis on the clinical implications of the study.

Reply) Thank you for your valuable opinion; we have condensed the Discussion.

Reviewer 2 Report

Comments and Suggestions for Authors

In this study by Namgung et al, the authors performed a cross-sectional analysis of data from the Korean National Health and Nutrition Examination to assess the associations among diet quality using the Korean Healthy Eating Inex (KHEI), sleep duration and obesity. They found that obesity was associated with insufficient sleep, but only in women with a KHEI less than their median. The paper is well written and results clearly presented. I have the following comments:

1. A Strobe diagram showing the included and excluded subjects would be helpful

2. What were the justifications for classifying some of the covariates. For example, occupation was dichotomized as working vs. jobless. Perhaps a more granular classification would be more informative such as student, part-time work, retired, etc. For physical activity, more than 2 classifications also might be useful.

3. Was there any attempt to analyze individuals sleeping more than the recommended amounts? Other studies have found adverse outcomes with longer sleep durations.

4. Why was a median split used to classify the KHEI as opposed to the mean or as a continuous variable?

5. The discussion seems a bit arduous and/or detailed. The descriptions of other relevant studies could be abbreviated.

Author Response

We sincerely appreciate the reviewer's insightful and constructive comments and suggestions. Please see our detailed responses in the attachment.

Round 2

Reviewer 1 Report

Comments and Suggestions for Authors

The authors have sufficiently answered to all comments